# Myxoid Endometriosis: An Entity That Can Cause Confusion with Malignant Entities

**DOI:** 10.3390/diagnostics13203176

**Published:** 2023-10-11

**Authors:** Saulo Mendoza-Ramirez, Claudia Mariana Hernández-Robles, Italú Velasco-Rueda, Alejandra Romero-Utrilla, Myrna Doris Arrecillas-Zamora, Eduardo Agustín-Godínez, Lourdes Lucía Morales-Jáuregui, Lázaro Ariel Ramírez-Balderrama, Marco Antonio Olvera-Olvera, Mario Murguia-Perez

**Affiliations:** 1Hospital General de Mexico “Dr. Eduardo Liceaga”, Mexico City 06720, Mexico; dr.saulomr.oncopato@gmail.com (S.M.-R.); alejandraromeroutrilla@gmail.com (A.R.-U.); myrna_doris@yahoo.com.mx (M.D.A.-Z.); 2Centro Médico ABC Observatorio, Mexico City 01120, Mexico; 3UMAE Hospital de Especialidades N° 1 Centro Médico Nacional Bajío, Instituto Mexicano del Seguro Social, Guanajuato 37328, Leon, Mexico; cmarianahr@gmail.com (C.M.H.-R.); dr.agustin.patologia@gmail.com (E.A.-G.); larb721201@gmail.com (L.A.R.-B.); 4Hospital General de Zona N°. 1 “Dr. Emilio Varela Luján”, Instituto Mexicano del Seguro Social, Zacatecas 98000, Zacatecas, Mexico; velascoitalu@gmail.com; 5Hospital General Regional N° 58, Instituto Mexicano del Seguro Social, Leon 37278, Guanajuato, Mexico; dra.lucia.morales@gmail.com; 6Hospital General de Zona con Medicina Familiar N° 21, Instituto Mexicano del Seguro Social, Leon 37380, Guanajuato, Mexico; macox7@gmail.com

**Keywords:** endometriosis, myxoid endometriosis, malignant neoplasms, histochemical stains, immunohistochemistry

## Abstract

Myxoid endometriosis, a rare entity, is part of the histological changes that can occur in endometriosis. Pathologists must know the histological guidelines for the morphological recognition of this entity, as well as the histochemical and immunohistochemical techniques that support diagnosis, and define the morphological characteristics of myxoid endometriosis. In the present work, we propose diagnostic guidelines and primary differential diagnoses using special histochemical techniques and immunohistochemical reactions to recognize this entity.

## 1. Introduction

The term “endometriosis” was first introduced by Sampson in 1927 based on the description of endometrial tissue in the myometrium by Rokitansky and in the rectovaginal septum by Cullen, who referred to this entity as adenomyoma and “hemorrhagic (chocolate) cysts in the ovaries”. Endometriosis is an inflammatory, chronic, benign, and estrogen-dependent disease that affects approximately 10% of women of reproductive age and 35–50% of women with dysmenorrhea and infertility [1]. The definition of endometriosis is histological and requires the identification of glands and stroma outside the uterus. These ectopic lesions are frequently found in the pelvic organs and peritoneum, and occasionally, in other body parts, such as the kidneys, bladder, lungs, and brain [2].

Endometriosis is a condition in which the endometrial tissue is composed of glandular epithelium and stroma found in anatomical sites outside the uterine cavity. The etiology, in most cases, is explained by the theory that the endometrium implants in the peritoneal cavity are the result of retrograde flow from the uterus to the uterine tubes. However, many diagnostic problems can arise due to abnormalities or absence of either glandular or stromal components, as well as discrete scant tissue in biopsies. The microscopic appearance of both the glandular and stromal components can be affected by hormonal changes, metaplastic changes, cytological atypia, hyperplasia, foamy or pigmented histiocytic infiltrate, fibrosis, elastosis, smooth muscle cell metaplasia, myxoid change, and decidual change. Therefore, the histological diagnosis of endometriosis can be a challenge for pathologists when any of the components show microscopic changes or when unusual anatomical sites are involved. Myxoid changes can easily be confused with mucinous malignancies if the pathologist is not familiar with the entity.

Myxoid endometriosis is a rare entity that is part of the histological changes that can occur in endometriosis. Pathologists must know the histological guidelines for the morphological recognition of this entity, as well as the histochemical and immunohistochemical techniques that will support the diagnosis.

## 2. Material and Methods

This study is a retrospective, observational, cross-sectional, and descriptive study. Pathology reports were obtained from the pathology files of two medical institutions, namely Hospital General de México “Dr. Eduardo Liceaga” and UMAE Hospital de Especialidades N° 1 Centro Médico Nacional Bajío, affiliated with the Instituto Mexicano del Seguro Social, with a diagnosis of endometriosis. The study covered a period of 6 years, from 1 January 2012 to 31 December 2021.

All results with a diagnosis of endometriosis were peer-reviewed by two oncology pathologists (M.M.-P., S.M.-R.) to confirm the initial diagnosis. Once confirmed, only cases that met the histological criteria for myxoid endometriosis, characterized by the presence of extracellular matrix with myxoid change, were included in the study. Cases that were incomplete and lacked paraffin blocks or slides or cases with issues in the histological technique procedure were excluded.

For each selected case, a paraffin block was chosen, and special histochemical techniques were performed. These techniques included periodic acid-Schiff staining (PAS) and Alcian blue staining (AA). Positive staining for the myxoid matrix and intracellular mucoid component was assessed. Additionally, immunohistochemical reactions were conducted for stromal cells (using CD10 antibody, Biocare^®^ (Pacheco, CA, USA), 1:100 dilution) and hormone receptors (estrogen receptors (ER) and progesterone receptors (PR) using Diagnostic Biosystem^®^ (Pleasanton, CA, USA) antibodies, both at 1:50 dilution). CD10 positivity was determined based on membranous expression, and for hormone receptors, nuclear expression with any intensity was considered (H-SCORE).

The data collected from these analyses was used to create a database in Excel. Descriptive statistics were employed to determine percentage frequencies, and the SPSS 21 (IBM^©^) (Armonk, NY, USA) program was used for the statistical analysis.

## 3. Results

A total of 776 cases of endometriosis were collected between the two institutions that participated in the study, in different anatomical sites (Table 1, Figure 1), of which 54 presented with a myxoid stroma >50% with respect to the cellularity of the samples and represented 6.95% of the total endometriosis examined, with the following locations: omentum (4/20), ileum (2/4), ovary (14/216), parametria (2/9), soft tissues (28/235), salpingus (3/77), and bladder (1/4). The clinical records of these patients were consulted, and the common denominator was that these women were in the surgical (41/57) or physiological (7/39) puerperium during 24 h postpartum to 6 months after said event. Macroscopically, two different lesions were observed: the first one, which is the most frequent, presents as a non-encapsulated lesion with pushing edges of a fibromyxoid appearance; the second and less frequent, is a well-defined lesion, partially or totally encapsulated, with a shiny cut surface, gelatinous in appearance, multilobed, and light brown to brown in color with focal areas of recent and old hemorrhage; these lobes are separated by fibrous septa (Figure 1). Histologically, it was found that in all the samples evaluated, at least 50% of myxoid stroma with some fine connective tissue septa with the proliferation of stromal cells (Figure 2A) presented the following four variants in their shape: (1) Epithelioid stromal cells of wide eosinophilic cytoplasm with round small nuclei (Figure 2B), (2) pseudolipoblast-type stromal cells with moderate cytoplasm, multivacuolated in their cytoplasm that can present eosinophils or clear cytoplasm, with central nuclei of granular chromatin (Figure 2C), (3) “pseudosignet ring” stromal cells with the nucleus and cytoplasm rejected to the periphery giving the described appearance (Figure 2D), and (4) immersed spindle cells with discrete nuclei (Figure 2E). Among these cells, the epithelial component was observed, with tubules ranging from small to large cystic dilations of the gland that present a layer of low cubic to cylindrical epithelium, generally with an atrophic appearance (Figure 2F). Hyaline globules were found in the glandular lumens (Figure 2G). The extracellular matrix showed a composition of glycosaminoglycans in 100% of the cases by staining present for AA (Figure 3A–C), but without staining for PAS; however, the latter stained on proteinaceous globular material in the lumen of the endometrial glands (Figure 3D). Immunohistochemistry for CD10 was positive in 100% of the stromal cells with a mild to moderate reaction in the cytoplasm and membrane with a decidual appearance (Figure 4A,B). The ER and PR showed nuclear expressions in 100% of cases, with greater intensity in the ER (Figure 4C) than PR (Figure 4D); both hormone receptors were expressed on the endometrial epithelial and stromal cells. The cases of myxoid endometriosis examined did not present cytological atypia, suggesting any precursor lesion of epithelial or stromal origin.

## 4. Discussion

### 4.1. General Aspects of Endometriosis

Endometriosis is a complex and enigmatic disease with an uncertain etiology. Several theories have attempted to explain its histogenesis, yet the exact cause remains unknown. Establishing an endometriotic implant requires specific conditions, including ectopic glands and stroma, adhesion of endometrial cells to the peritoneum, invasion of the mesothelium, and maintenance and growth of ectopic tissue. These features share similarities with processes reported in neoplasms and involve various biological reactions. Likely, both intrinsic factors within the ectopic endometrium and immunological alterations in the host play crucial roles in the development of endometriosis [2]. The retrograde theory, the oldest among the proposed explanations, suggests that endometriosis occurs due to the retrograde flow of endometrial cells and detached debris passing through the uterine tubes into the pelvic cavity during menstruation [1,3]. Conversely, the metaplastic theory proposes that endometriosis arises from metaplasia of specialized cells of the visceral mesothelium and abdominal peritoneum. This theory suggests that hormonal and immunological factors stimulate the transformation of normal cells and tissues in the peritoneum into endometrial-like tissue, which may explain the incidence of endometriosis in prepubertal girls. However, the estrogen stimulus is not present in these patients, making this condition potentially distinct from endometriosis occurring in women of reproductive age. Evidence of ectopic endometrial tissue in female fetuses further supports the theory that endometriosis results from defects in embryogenesis, with residual embryological cells from the Wolffian or Müllerian ducts persisting and evolving into endometriotic lesions that respond to estrogen stimulation. 

Some scholars have proposed endogenous, biochemical, and immunological factors as potential causes acting as inducers of differentiation in ectopic endometriotic tissue [1]. The support for the theory of a non-endometrial origin of endometriosis arises from histologically confirmed clinical cases in patients without menstrual endometrium, such as patients with Rokitansky-Kuster-Hauser syndrome and men with prostatic carcinoma undergoing treatment with doses high estrogen [4].

The Cancer Genome Atlas (TCGA) Research Network, in 2013, changed the view of the classification of endometrial cancer (EC), with the integration of molecular characterization, addressing numerous limitations in risk stratification that, for decades, was based solely on tumor grade and histotype, depth of myometrial invasion, and cervical and adnexal involvement. Adding new variables, such as mutations and somatic variations, in copy number, genome and exome sequencing, and determining microsatellite instability (MSI), has helped to divide the new CE classification into 4 groups with different prognoses in terms of free survival of specific progression and risk of recurrence, namely Polymerase epsilon (POLE) ultramutated, MSI ultramutated, copy number (CN) low, and CN high [5]. Although oncologists are increasingly using this classification to determine the type of treatment, despite the fact that validation is lacking, only advanced or metastatic stages could benefit from targeted adjuvant therapies based on molecular alterations, particularly considering advanced MSI-H/MMR (dMMR) deficiency. Numerous studies have evaluated the efficacy of monoclonal antibody therapy directed against immune checkpoint-associated proteins expressed at high levels within the tumor microenvironment and making tumor cells susceptible to the immune system response [6]. The molecular characteristics of endometriosis comprise a hormone-dependent inflammatory condition (estrogen dependence, progesterone resistance) with an (epi)genetic predisposition most likely driven by cells with plasticity. Recently discovered biological concepts (e.g., exosomes and miRNAs) are also relevant to the pathogenesis of endometriosis. Exosome-mediated intercellular communication is a potentially new mechanistic tool to orchestrate cell fate, for example, modulating signaling pathways. A challenge in endometriosis research is the evaluation of nonsteroidal signaling pathways as targets for new therapies [7]. Molecular studies of endometriosis have reported significant findings based on the Wnt-B-catenin signaling pathway and its relationship with the levels of vascular endothelial growth factor (VEGF) and E2, whose association triggers decreased levels of GSK3β and E-cadherin and increased levels of nuclear β-catenin in endometriotic lesions when compared with uterine endometrial tissue. Such data are consistent with the idea that decreased GSK3β stabilizes β-catenin, which exhibits elevated expression and presence in the nucleus. This in turn should enhance the expression of β-catenin target genes of which VEGF is required for angiogenesis. Why endometriotic lesions exhibit a decrease in E-cadherin mRNA as compared with endometrial tissue remains unclear. A rather trivial explanation for this observation is that ectopic endometrium contains more stromal than epithelial cells. However, E-cadherin appears dysregulated in several cells of ectopic lesions as previously shown by our group and others [8,9,10]. This indicates that, despite the relationship between endometriosis and hormonal factors, in the molecular aspect, no significant relationships have been found to support its association with the development of EC, although in the protocols of the College of American Pathologists (CAP), information is requested in the preparation of anatomopathological reports [11].

Genetic factors have been implicated in the development of endometriosis, supported by familial case reports, a higher risk of endometriosis in patients with affected first-degree relatives, and the observation of concordance of endometriosis in twins. Numerous studies have identified genetic polymorphisms as contributing factors to the development of endometriosis. The disease appears to have a polygenetic mode of inheritance involving multiple loci and chromosomal regions associated with specific endometriosis phenotypes. Both inherited and acquired genetic factors may predispose women to have an impaired response to discarding endometrial tissue that adheres to the peritoneal epithelium. Genes associated with detoxification enzymes and ER polymorphisms and genes of the innate immune system are among those implicated in the pathogenesis of endometriosis. Additionally, epigenetic changes such as DNA methylation, demethylation, and modifications in the histone code may also contribute to the development of the disease [1,4]. Table 2 summarizes these aspects.

### 4.2. Myxoid Change

The term “myxoid” was introduced by Rudolph Virchow in 1858 to describe soft tissue tumors resembling the umbilical cord’s structure. The concept of myxoid tumors led to the recognition of various new tumors, including myxadenoma, myxochondroma, myxofibroma, and myxoneuroma. Today, myxoid changes are described in areas identified in both benign and malignant neoplasms, as well as in non-neoplastic (reactive) lesions [12]. Advancements in the study of the myxoid extracellular matrix introduced the alcian blue histochemical staining in 1950, which allowed for the distinction between different glycosaminoglycans (GAGs) in tissues. GAGs are macromolecules present in the pericellular and extracellular matrix and form proteoglycans once covalently attached to specific nuclear proteins [13].

From a biochemical perspective, the term “myxoid” encompasses various proteins and other macromolecules with different functions. When encountering a myxoid lesion, an accurate diagnosis primarily relies on careful histopathological evaluation based on routine criteria such as tumor demarcation, growth pattern, vascular pattern, and nuclear atypia. Immunohistochemistry may be useful for differential diagnosis in certain cases, although it is not always decisive. In instances of diagnostic difficulty, molecular cytogenetics plays a crucial role. Beyond its academic significance, understanding the role of glycosaminoglycans and proteoglycans in the biology of myxoid lesions is essential, as the myxoid morphology of the extracellular matrix can be found in non-neoplastic reactive processes, benign, and malignant tumors alike [12]. 

### 4.3. Myxoid Changes in Uterus

Myxoid lesions occurring in the uterus are rare, and most of them represent smooth muscle myxoid tumors, either benign or malignant, as well as a variant of endometrial stromal neoplasia. While a myxoid stroma characterizes some uterine lesions as a defining feature, myxoid changes may also arise due to degenerative processes or be associated with the use of medications, such as progestogens or progestins. Progestogens are synthetic analogs of progesterone commonly used in the management of abnormal uterine bleeding, endometriosis, contraception, and as protection for patients undergoing tamoxifen treatment. Patients using these medications may exhibit morphological changes in the endometrial stroma, such as patchy edema with intercellular vacuolation, decidualization, hemorrhage, and myxoid change in the stroma. Pathologists must be aware of these morphological characteristics associated with progestogen use to avoid misdiagnosis (Table 3 summarizes findings in endometriosis associated with diagnostic pitfalls) [12,14].

Occasionally, the endometrial stroma can undergo marked myxoid change, with the stromal cells separated by acellular mucin, sometimes forming “pools” of mucin. The mucin in these cases is stromal, not epithelial, and stains positively with toluidine blue or alcian blue (pH 2.5) but not with periodic acid-Schiff (PAS). Some cases have occurred during pregnancy or puerperium, suggesting a potential association with hormonal activity. The presence of marked decidual transformation of the endometriotic stroma may further complicate the histological interpretation in pregnant patients [13,14]. Awareness of these changes and the presence of endometrial glands, even if occasionally atrophic, can aid in the correct diagnosis.

Endometriosis with myxoid change in the stroma is rare and not commonly known among histopathologists, as it is not extensively described in pathology texts. Less than ten cases of endometriosis with myxoid change have been reported, with some exhibiting histological similarity to metastatic adenocarcinoma, pseudomyxoma peritonei, and even signet-ring cell carcinoma-like features [3]. 

### 4.4. Diagnostic Problems in Endometriosis

Many diagnostic problems can arise due to alterations or absence of any of the components, glandular or stromal, and scant tissue in tiny biopsies. The microscopic appearance of the glandular component can be affected by hormonal changes, metaplastic changes, cytologic atypia, and hyperplasia. In some cases, the endometriotic glands are scattered or even absent (stromal endometriosis). The stromal component can be obscured by foamy or pigmented histiocytic infiltrate, fibrosis, elastosis, smooth muscle cell metaplasia, myxoid change, and decidual changes. The histological diagnosis of endometriosis can also be challenging for pathologists when involving rare anatomical sites, considering the five most problematic sites namely the ovary, cervix, vagina, uterine tubes, and intestine, including the critical distinction between an endometrioid carcinoma, arising from colonic endometriosis, from primary colonic adenocarcinoma [4,9]. 

The histological diagnosis of endometriosis is usually simple and based on the typical presence of both components, endometriotic glands, and stroma, although it can be diagnosed when only one of the components is present. Similar to normal and neoplastic endometrial stroma, endometriotic stromal cells are immunoreactive for CD10, which may aid in diagnosis in situations such as sparse tissue or absence of the glandular component. It is important to define the following two concepts that can also generate diagnostic problems:Atypical endometrioid-type endometriosis: a lesion with intermediate characteristics between endometriosis and endometrioid-type adenocarcinoma. It may resemble an epithelial endometrial neoplasm of the endometrium by having prominent architecture, increased glandular density, and various degrees of nuclear atypia. Endometrioid adenocarcinoma arising from endometriosis shows loss of function of PTEN (21%), KRAS (20%), β-catenin (25%), and PIK3CA (46%).Atypical endometriosis of the clear cell type: contains cells with nuclear atypia and vacuolated or foamy cytoplasm and is a precursor lesion of clear cell adenocarcinoma. Atypical epithelium may be exfoliative, studded, or flattened. There may also be complex glandular formations of exophytic epithelium and dense aggregates. It has been shown to have mutations in PIK3CA and loss of the ARID1A protein, as in clear cell carcinoma. Diagnosis requires the presence of marked nuclear pleomorphism and hyperchromasia.

The distinction between these two concepts is challenging since they tend to overlap and show a tremendous histological variety. Atypical histology of both endometrioid and clear cell types of endometriosis often becomes apparent only when the disease has progressed to carcinoma. Another complication is that, to some degree, it is common for cells to show frank atypia, exfoliation, and cell translocation in areas with degenerative changes [15].

### 4.5. Diagnostic Problems Related to the Endometrial Stromal Component

Microscopic alterations in the typical appearance of endometriosis that can hinder its diagnosis occur in both the glandular and stromal components, which is more common. Although these changes can mask the typical stromal component, their presence should increase the diagnostic suspicion of the endometrial origin of the lesion. In these cases, the endometrial stroma, if present at all, may be very subtle and confined to a faint or discontinuous periglandular area or delineating an endometriotic cyst. In the latter case, if the epithelial cells are denuded, the stromal cells may rest directly on the cyst lumen. 

If the stromal origin of the cells is in doubt, immunohistochemical reactions with CD10 [4] can be used.

### 4.6. Differential Diagnoses

#### 4.6.1. Myxoid Chondrosarcoma

Myxoid chondrosarcoma, a rare tumor that corresponds to 2.5% of all soft tissue tumors, was first described by Enzinger and Shiraki in 1972 and mainly affects middle-aged men, manifesting as a painless, slow-growing tumor. The frequently affected anatomical sites are the extremities, but can also affect the neck, orbit, and peritoneum. Grossly, it is a well-circumscribed, pseudoencapsulated, lobulated, light-brown to gray tumor with a shiny, whitish-grey to light-brown surface on a section containing mucoid material and may present with cystic or hemorrhagic degeneration. Microscopically, it shows a multilobulated neoplasm made up of polygonal to spindle-shaped malignant cells, forming nests and cords on a myxoid matrix, separated by fibrous septa. Despite being cartilaginous, it is strongly immunoreactive for vimentin and variable for S-100 [11]. 

#### 4.6.2. Myxoma

Myxomas, a heterogeneous group of benign soft tissue tumors, were first described by Virchow in 1871. They originate from primitive mesenchyme, resembling the structure of the myxoid connective tissue of the umbilical cord. Most are deep lesions and occur in the skin, subcutaneous tissue, genitourinary tract, gastrointestinal tract, and solid organs such as the liver, spleen, and even the parotid gland [14]. Myxomas commonly affect patients between 40–70 years of age, with a predilection for women of 57%; they present as a slow-growing tumor that may or may not be painful [15]. Histologically, they are characterized by scattered, stellar bipolar cells among a vascularized myxoid matrix and may present hypercellular and hypocellular regions with fibrous areas, few mitoses, and bland nuclear chromatin [16].

#### 4.6.3. Myxoid Liposarcoma

The myxoid histological variant corresponds to 15–20% of all liposarcomas. It is more commonly found in young patients with a peak incidence between the fourth and fifth decades of life and arises mainly from the soft tissues in the extremities, with uncommon anatomical features including the head, neck, subcutaneous tissue, and thorax.

Histologically, it shows a uniform mixture of oval cells and signet-ring cells corresponding to lipoblasts on a background of myxoid stroma and a very prominent arborizing capillary vasculature [16]. 

#### 4.6.4. Myxofibrosarcoma

Formerly known as myxoid malignant fibrous histiocytoma, myxofibrosarcoma is considered one of the most common fibroblastic sarcomas in the elderly, mainly affecting patients between 60 and 80 years of age. It is a unique subtype of soft tissue tumors, characterized by a diffuse infiltrating pattern, represents approximately 5% of the diagnoses of sarcomas in soft tissues, and was originally described in 1977 as a multilobulated tumor, with myxoid characteristics on the cut surface. Microscopically, a hypocellular myxoid neoplasm is observed, with myxoid areas containing spindle cells, sometimes pleomorphic. The vascular network is evident, and in the more cellular areas, mitosis is common. Sometimes, myxoid areas contain “pseudo-lipoblasts” and epithelioid cells that grow diffusely, conferring a more aggressive behavior of the neoplasia. There are no specific immunohistochemical reagents for this tumor; it can be positive for vimentin and CD34 and negative for S100 [17].

#### 4.6.5. Myxoid Leiomyosarcoma

Uterine myxoid leiomyosarcomas are exceptionally rare and were first described by King et al. in 1982. In this study, 6 cases macroscopically characterized by a gelatinous appearance were reported. In the microscopy study, they described neoplastic cells surrounded by an abundant amount of myxoid material with a low mitotic index (0–2 mitoses per high power field). The age range of presentation is from 47 to 68 years and manifests as a vaginal or pelvic tumor that produces abnormal bleeding. Although myxoid degeneration in leiomyomas has an incidence of 13%, this finding is very rare in leiomyosarcomas. Microscopically, myxoid leiomyosarcoma is characterized by spindle cells with smooth muscle characteristics, surrounded by a profuse myxoid stroma. Diagnosis requires the presence of mitosis, nuclear atypia, and tumor necrosis. The most important indicator of malignancy is the mitotic index, hence it is the guideline that distinguishes leiomyosarcomas from cellular leiomyomas. Although myxoid leiomyosarcomas generally have a low mitotic index, they have a similar malignant potential to classical leiomyosarcoma; in these cases, myxoid stroma may be a determining factor. Inflammatory myofibroblastic myxoid tumors can be differentiated from myxoid leiomyosarcoma by the presence of lymphoplasmacytic inflammatory infiltrate and immunoreactivity for ALK-1 [18]. 

### 4.7. Scope and Limitations of the Study

The limitations of this study were that it was carried out on retrospective histopathological material, directly identifying the histopathological findings proposed in the Section 3. This does not suggest that the data are deficient; on the contrary, in a huge sample of histopathological material, we could identify a rare change that is easily confused with malignant neoplasms. The scope of the study was to demonstrate to pathology residents and physicians who frequently observe gynecological specimens how to identify these changes and share them through diagnostic support networks, or, where appropriate, support and publish them for greater knowledge. These changes are very frequently observed, but due to a misdiagnosis, patients are considered “cured” of their malignant tumors but continue to have endometriosis symptoms. This study also provides an opportunity for collaboration between gynecologists and pathologists to address cases that are difficult to diagnose.

### 4.8. Utility of Immunohistochemistry 

Immunohistochemistry is a powerful tool that allows for differentiating between different histotypes. However, for individuals who are poorly trained in this technique or those with little experience, immunohistochemistry can cause diagnostic confusion, especially when the necessary clinical data are not available. When approaching a tumor like the one presented in Figure 1, the lack of suspicion of myxoid endometriosis, ignoring the histological criteria that we proposed, can draw a thin line between benignity or malignancy diagnosis. Immunohistochemistry should only be used in the case of a suspected diagnosis, and only pathologists can decide on its use, which is why, although the study and review are of general utility, the approach directed to pathologists is vital for the recognition of this entity.

## 5. Conclusions

Endometriosis presents a wide range of histopathological aspects, one of which we propose as an infrequent variant that can be easily confused with malignant tumors (liposarcoma, myxofibrosarcoma, mucinous carcinoma, etc.). Even among pathologists with extensive experience, we note the diagnostic difficulty. Immunohistochemistry must confirm diagnostic suspicion, showing constant expression of CD10, ER, and PR in endometriosis. When the endometriosis presents <50% of myxoid background, we propose the term “endometriosis with myxoid changes”. On the other hand, if the myxoid background is >50% of the sample, we propose the term “myxoid endometriosis”. This separation is due to the more significant myxoid stromal component; the more heterogeneous the histological findings the more the potential differential diagnoses.

## Data Availability

Not applicable.

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
