# Peer review of "Myxoid Endometriosis: An Entity That Can Cause Confusion with Malignant Entities"

_diagnostics, 2023, doi:10.3390/diagnostics13203176_

Round 1
Reviewer 1 Report
Dear author's
I was pleased to review your manuscript and i have the following comments:
First of all the subject of your article does not meet the SI Theme.
The article described a multitude of endometriosis lesions. Please explain what new information brings your article in the literature.
There are multiple paragraphs without references. Please revise.
The images are interesting and qualitative.
The conclusion and recommendation are not clear: The IHC is necessary in all cases or only in those with unclear diagnosis? Please explain
English and punctuation edits are required.
References does not meet the journal criteria - please see instruction for authors.
Author Response
I appreciate the review, comments and areas of opportunity to improve the manuscript, I respond to each of the points:
- I respectfully disagree with your point of view. The theme of the manuscript is that myxoid endometriosis simulates malignant entities, and what we do in the review of the work is to explain morphological and immunophenotype criteria of the characteristics of this entity, in comparison with morphological and immunophenotype criteria of malignant entities.
- The topic of the collection is Trends and Controversies in Global Gynecologic Oncology: Diagnosis and Management. I understand that we do not provide comments on management, but that endometriosis can simulate malignant neoplasms, and that a more extensive study must be done regarding the patients' symptoms, as well as additional studies if required, is a point to be made. favor, especially for pathologists who are the ones who have this daily battle. The information it provides are recommendations for the suspected diagnosis, as well as discussing certain points such as myxoid change, changes related to the epithelial and stromal components, among other things. It is extremely important for pathologists, as morphologists and evaluators of special studies, to be able to take these changes into account, just when there are no manuscripts that talk between endometriosis and its simulations with malignant neoplasms.
- Some paragraphs are not referenced because their reference is the consecutive one of the following paragraph, we did not want to be repetitive. But we have corrected and in some cases we omitted the paragraph and merged it, and in the other case we added the corresponding reference.
- The images are interesting and qualitative = the images are completely representative of the situations in which the pathologist might have a confusion, especially when there is extensive myxoid change.
- Immunohistochemistry is recommended when the pathologist suspects that it is myxoid endometriosis or with myxoid change, in women, and, above all, when there are reference clinical data. The biggest challenge is for the pathologist to know the entity, and based on this, to make a precise diagnosis from the hematoxylin-eosin study, to relying on immunohistochemistry and ruling out the idea of a malignant neoplasm. An annex was added specifying this reflection, which is specifically aimed at pathologists.
- The manuscript was reviewed by native English speaking American medical colleagues. However, the manuscript was resubmitted to the same American pathology colleagues with native and medical English, and readability and typography were improved according to their recommendations.
- References are in Vancouver format, and the Mendeley platform was used to add them to the text. In the instructions for the authors, they recommend some software, but do not mention this as mandatory. We have reviewed the format of the references, and we consider them to be adequate, obviously respecting your opinion.
- The modified text, as well as the new text added with the respective references, the reorganization of the neoplastic entities to be ruled out, and a new item on the scope and limitations of the study, as well as the use of immunohistochemistry, were highlighted in yellow.
Reviewer 2 Report
I read with great interest the manuscript, which falls within the aim of this Journal and offers a high-quality overview of the topic.
Although the manuscript can be considered already of high quality, I would suggest taking into account the following minor recommendations:
- I suggest another language revision round to correct a few typos and improve readability.
- I find it interesting to include a reference to the molecular mechanisms of endometrial disease that can help clinicians during the diagnostic path and influence the therapeutic strategy (see PMID: 36979434).
- Inclusion/exclusion criteria should be better clarified by extending their description.
- The authors have not adequately highlighted the strengths and limitations of their study. I suggest better specifying these points.
The whole text should be corrected by a native English speaker in order to make the work clearer and more readable.
Author Response
I appreciate the review, comments and areas of opportunity to improve the manuscript, I respond to each of the points:
- The manuscript was resubmitted to the same American pathology colleagues with native and medical English, and legibility and typography were improved according to their recommendations.
- It is certainly a difficult morphological diagnosis and we emphasize the adequate histopathological approach and demonstration of the disease. For this reason, we decided to add, in addition to the recommended reference, some others, as well as the different molecular mechanisms between endometrial carcinoma. and endometriosis.
- The inclusion and exclusion criteria were restructured in such a way that they can be better understood.
- The strengths are that it is an original work in which we try to demonstrate an entity little described or even only mentioned in the literature that is easily confused with other neoplasms, both malignant and benign, and that its correct diagnosis is morphological and that studies are rarely needed. immunohistochemistry. Being an attempt to describe a relatively new entity with few tools, the limitation is the dissemination and knowledge of this entity by the majority of pathologists.
- The modified text, as well as the new text added with the respective references, the reorganization of the neoplastic entities to be ruled out, and a new item on the scope and limitations of the study, as well as the use of immunohistochemistry, they were highlighted in yellow.
Reviewer 3 Report
To The Chief Editor
Diagnosis
The manuscript “Myxoid endometriosis: an entity that can cause confusion with malignant entities” is well written and very precise where the author explains endometriosis based on histological changes in the endometriosis. Before this paper can be accepted for publication some points need to be addressed.
Major issues.
1. In this paper the author discussed only the histological changes in endometriosis. The manuscript needs more experiments to be performed to validate the findings.
2. The presentation of data in the manuscript is poor and it is not organized. The author should discuss the result of histological findings in the manuscript.
3 The manuscript needs more references to support the statement of the manuscript.
4. The author should discuss the limitations and challenges of this study.
Efforts are needed to improve the language of the manuscript.
Author Response
I appreciate the review, comments and areas of opportunity to improve the manuscript, I respond to each of the points:
- Certainly only histological changes are analyzed with the help of histochemistry and supported by immunohistochemistry, we consider that for diagnostic purposes; no further special studies such as electron microscopy or molecular biology are necessary.
- The information was reorganized in a more user-friendly manner and emphasis was placed on these histological findings.
- Unfortunately there is not much literature that describes this entity as such, there are case reports, but there are no series dedicated to this myxoid change in particular and even less with the problem we face of making the differential diagnosis with other entities, as it is a description original of a little known condition. We consider that this entity is new, it must be described, but be based on what currently exists. However, we do add new references regarding the molecular differences between endometriosis and endometrial carcinoma.
- The limitations are basically that it is not an entity known to the pathologist, thus the idea was born, when we saw such discordance in the diagnosis, from benign to malignant entities and even carcinomas with signet ring cells, we undertook the task of look for the characteristics as descriptive findings with the help of histochemistry and immunohistochemistry. The biggest challenge is for the pathologist to know it and make an accurate diagnosis from the study with hematoxylin-eosin, to relying on immunohistochemistry and ruling out the idea of a malignant neoplasm. An annex was added specifying this reflection, which is specifically aimed at pathologists.
- The modified text, as well as the new text added with the respective references, the reorganization of the neoplastic entities to be ruled out, and a new item on the scope and limitations of the study, as well as the use of immunohistochemistry, they were highlighted in yellow.